# Active Pregnancy: A Physical Exercise Program Promoting Fitness and Health during Pregnancy—Development and Validation of a Complex Intervention

**DOI:** 10.3390/ijerph19084902

**Published:** 2022-04-18

**Authors:** Rita Santos-Rocha, Marta Fernandes de Carvalho, Joana Prior de Freitas, Jennifer Wegrzyk, Anna Szumilewicz

**Affiliations:** 1ESDRM Sport Sciences School of Rio Maior, Polytechnic Institute of Santarém, 2040-413 Rio Maior, Portugal; marta.fernandes@go-fit.pt (M.F.d.C.); joana.pfreitas150@gmail.com (J.P.d.F.); 2CIPER Interdisciplinary Centre for the Study of Human Performance, Faculty of Human Kinetics (FMH), University of Lisbon, 1499-002 Cruz Quebrada, Portugal; 3HESAV School of Health Sciences, HES-SO University of Applied Sciences and Arts Western Switzerland, 1011 Lausanne, Switzerland; jennifer.masset@hesav.ch; 4Department of Fitness, Gdansk University of Physical Education and Sport, 80-336 Gdansk, Poland; anna.szumilewicz@awf.gda.pl

**Keywords:** physical activity, exercise, health, fitness, pregnancy, complex intervention, CREDECI2, reproducibility

## Abstract

Physical activity during pregnancy is a public health issue. In the view of reproducibility and the successful implementation of exercise interventions, reporting the quality of such study design must be ensured. The objective of this study was to develop and validate a physical exercise program promoting fitness and health during pregnancy. A qualitative methodological study was carried out. For the description of the exercise program, the Consensus on Exercise Reporting Template (CERT) was used. For the validation of the program, the revised guideline of the Criteria for Reporting the Development and Evaluation of Complex Interventions in Health Care (CReDECI2) was followed and went through three stages of development, piloting, and evaluation. The customizable exercise program was designed and validated by exercise and health specialists based on evidence-based, international recommendations and supported by different educational tools to be implemented by qualified exercise professionals in health and fitness settings. A 12-week testing intervention addressing a group of 29 pregnant women was carried out. The program’s feasibility was subsequently evaluated by all the pregnant women. The CReDECI2 process guides practitioners and researchers in developing and evaluating complex educational interventions. The presented intervention may assist exercise specialists, health professionals, and researchers in planning, promoting, and implementing a prenatal exercise program.

## 1. Introduction

Physical activity and exercise programs improve fitness and well-being across the lifespan of the general population and are an effective means to prevent and manage acute and chronic health conditions [1]. The World Health Organization (WHO) recommends that “women who, before pregnancy, habitually engaged in vigorous-intensity aerobic activity or who were physically active, can continue these activities during pregnancy and the postpartum period” [2]. Several systematic reviews support the promotion of moderate-to-vigorous prenatal physical activity for maternal health benefits [3,4,5,6,7,8,9,10,11,12,13,14]. Indeed, maternal well-being and quality of life are improved when women participate in regular group exercises during pregnancy, as shown by the large, randomized control trials (RCT) led by Stafne et al. [15], Ruiz et al. [16], Haakstad et al. [17], Sagedal et al. [18], and Barakat et al. [19,20].

Encouraging pregnant women to engage in exercise programs is crucial in managing their weight gain and maintaining a healthy lifestyle [21,22]. Mullins et al. [23] conducted a systematic review on the adherence of postpartum women to supervised and unsupervised exercise interventions as the primary outcome. Although the interventions were highly variable, the loss to follow-up was higher and adherence to the intervention was lower for unsupervised versus supervised studies [23]. No such studies were found so far that included pregnant women.

Yet, physical activity and exercise interventions lack standardized methodologies in the development, delivery, and assessment of such programs due to their complex, regarding the different interacting components, interprofessional and intervention contexts [3,4,24,25].

According to the Medical Research Council (MRC), a complex intervention is described as an intervention that contains several interacting components that can act either independently or interdependently [26,27,28]. Educational and behavior change interventions in healthcare can be considered as complex interventions and include adapted physical exercise programs tailored to a specific population and setting and addressing the safety considerations [29]. In 2021, the MRC published an updated framework for the development and evaluation of complex interventions [30]. In 2012, the original framework (published by the MRC in 2000) was developed and applied to the contexts of healthcare [31], and the revised guideline of Criteria for Reporting the Development and Evaluation of Complex Interventions in Health Care (CReDECI2) was published by the same authors in 2015 [28]. To date, only one exercise intervention has been reported according to the CReDECI2 by Santos-Rocha et al. in 2019 [32].

The Consensus on Exercise Reporting Template (CERT) by Slade et al. [33] is a template that can be used when constructing, submitting, reviewing, and publishing articles, including systematic reviews [34]. It has shown good rater agreement and can comprehensively evaluate the reporting of exercise interventions [33].

There are updated international guidelines for exercising during pregnancy and postpartum [35,36,37,38,39,40,41,42,43,44,45] that have been discussed by Evenson et al. [46] and Szumilewicz et al. [47]. However, these guidelines lack specific and structured methodological guidelines to concretely design and implement effective and safe exercise programs [48]. Moreover, we identified a gap in the publications providing evidence-based guidance for the description of the exercise interventions needed to develop and validate well-defined and replicable exercise protocols. Therefore, the aim of this study was to develop and validate a physical exercise program “Active Pregnancy”, promoting fitness and health during pregnancy.

## 2. Materials and Methods

### 2.1. Design

A qualitative methodological study was carried out.

### 2.2. Participants

Fifty participants were involved in the validation process, including eleven exercise specialists with PhD, MSc, or BSc in exercise sciences, working with pregnant women; three medical doctors (obstetrics/gynecologist); seven healthcare specialists (i.e., midwives, physical therapists, and a nutritionist) with PhD, MSc, or BSc in health sciences also working with pregnant women; and 29 pregnant women participating in the pilot intervention.

### 2.3. Instruments

The revised guideline CReDECI2 by Möhler et al. [28] was followed. The CReDECI2 guideline comprises 13 items for the stages of development, piloting, and evaluation and includes examples of real studies for each item [28].

The design of the prenatal exercise intervention was described using the 16 items internationally endorsed by the CERT by Slade et al. [33]. This description was included in item 2 of the CReDECI2.

A report on the pilot intervention experience of participation in an exercise program during pregnancy, including satisfaction, adherence, and adverse events, was conducted in accordance with Haakstad et al. [49].

### 2.4. Procedures

#### 2.4.1. Workshop

Prior to the pilot intervention, exercise specialists and healthcare professionals were invited to attend a one-day educational workshop in which the exercise program was presented, and feedback was collected to improve the design in accordance with the contextual barriers and facilitators. The features of this workshop are described in detail elsewhere [50]. The educational contents addressed in the workshop are published elsewhere [51].

#### 2.4.2. Exercise Program

The exercise program was designed through the three stages proposed by Möhler et al. [28]: development, piloting, and evaluation. The physical exercise program for pregnant women was designed by exercise specialists in partnership with health professionals. The plan includes a variety of exercises to promote cardiorespiratory fitness, posture, strength, flexibility, balance, and pelvic floor muscle training. A portfolio of exercises in digital format was built with links to online resources (videos on YouTube demonstrating the exercises) [52,53]. The dissemination process, recruitment of the target population and implementation of the pilot study in the gym were planned.

#### 2.4.3. Pilot Intervention

An in-person pilot intervention of 12 sessions over 4 weeks was delivered between October 2019 and February 2020 in Lisbon, Portugal. The pilot exercise intervention was carried out by two exercise physiologists, qualified prenatal exercise specialists, that previously attended the workshop. All pregnant women were examined by a gynecologist before being included in the pilot intervention. They were recruited by the gynecologists and the exercise physiologists who validated the designed exercise program. Inclusion criteria for the intervention group were: pregnant women 18–45 years of age without medical contraindications for the practice of physical exercise. Exclusion criteria were: medical contraindications for physical exercise. Participants were informed about the objectives and the characteristics of the intervention, and informed consent was obtained. Twenty-nine healthy pregnant women with singleton pregnancies between 20 and 25 weeks of gestation, without complications, with a mean age of 34 ± 5 years, physically active and with a healthy body mass index before pregnancy participated in the pilot intervention. No adverse events were reported during the pilot intervention.

After the pilot intervention, the pregnant women were asked if they were more motivated when carrying out a group activity, if they felt a greater commitment, and about the effects they felt on their physical fitness and health benefits.

### 2.5. Ethical Considerations

Healthy, pregnant women were invited to participate in the pilot intervention free of charge. An informed consent document was signed prior to participation. The women were informed about the objectives, the nature of the study, the potential benefits, the participation requirements, and their right to withdraw from the study at any time without any consequences. All educational materials produced by the research team were made available to the participants and partners. The study was conducted in accordance with the Helsinki Declaration. This study is part of the study protocol that was approved by the Ethics Committee of the Polytechnic Institute of Santarém, Portugal (approval number 9-2021-ESDRM).

## 3. Results

### 3.1. First Stage: Development

Item 1—Description of the intervention’s underlying theoretical basis.

Official guidelines published by national and international obstetrics, gynecology, or sports medicine institutions are a trustworthy and comprehensive source of information in terms of the safety and health benefits of exercise during pregnancy and should be fostered by health and exercise professionals [46]. The recently published official recommendations on physical activity during pregnancy were accessed, as follows, and further discussed by Szumilewicz et al. [47]:SMA—Sport Medicine Australia 2017 [35]U.S. DHHS—U.S. Department of Health and Human Services 2018 [36]IOC—International Olympic Committee 2018 [37]SOGC—Society of Obstetricians and Gynaecologists of Canada/CSEP—Canadian Society for Exercise Physiology 2018 [38]EIM/ACSM—Exercise is Medicine/American College of Sports Medicine 2019 [39]WHO—World Health Organization 2020 [2]RANZCOG—The Royal Australian and New Zealand College of Obstetricians and Gynaecologists 2020 [40]ACOG—American College of Obstetricians and Gynecologists 2020 [41]ACSM—American College of Sports Medicine 2020 [42]NHS—National Health Service (United Kingdom) 2020 [43]AGDH—Australian Government. Department of Health 2020 [44]SBC—Brazilian Society of Cardiology 2021 [45]

Moreover, recent systematic reviews have shown moderate to strong evidence on the effectiveness of (moderate to vigorous intensity) physical activity of:Improved physical activity level [24]Improved maternal cardiorespiratory fitness [11,12,13,25,54,55]Reduced resting heart rate and blood pressure [25]Reduced risk of excessive gestational weight gain [11,12,13,14,24,55]Reduced risk of gestational hypertensive disorders overall and gestational hypertension [6,7,11,12,13,14]Prevention and treatment of gestational diabetes mellitus combined with dietary adaptation [5,7,11,12,13,14,56,57,58]Prevention of urinary incontinence [10,11,14,55]Reduced cesarean delivery [59,60]Prevention and treatment of low back, pelvic girdle, and lumbopelvic pain [9,14,24,61,62]Prevention of antenatal and post birth depression and anxiety [8,11,14,63,64,65]Improved well-being and quality of life [24]Positive impact on offspring health outcomes [11,12,13,14,60], i.e., reduced excessive birth weight and presumably decreased risk of miscarriage [60].

In the last three decades, an increasing amount of scientific evidence has proven the positive effects of prenatal physical activity on maternal and fetal health, as well as on pregnancy outcomes. Yet, insufficient levels of physical activity have been seen in pregnant women worldwide [66]. Thus, physical inactivity during pregnancy is a significant public health issue due to its prevalence and association with adverse pregnancy and birth outcomes, as well as the short- and long-term risks for several chronic diseases for the mother and child [67].

Item 2—Description of all intervention components, including the reasons for their selection, as well as their aims/essential functions.

The prenatal exercise program was described using the 16 items internationally endorsed by the CERT, as follows [33]:

Item 2.1—Type of exercise equipment: The prenatal gym-based exercise program requires basic equipment, such as a bench step, rubber bands, free weights, water bottles, mats, sticks, bars, fitballs, and softballs, as illustrated in the e-book [52] and YouTube videos [53]: https://www.youtube.com/channel/UC0Vyookwc0mcQ5T70imtoNA/playlists (accessed on 15 March 2022).

This equipment is used to increase the relative load in the movement directions described for each exercise. Each exercise includes specific instructions regarding the setup of the exercise equipment, such as how to perform the proper technique.

Item 2.2—Qualifications, expertise, and/or training of the exercise professionals: The program was delivered by qualified and certified exercise physiologists with a degree (bachelor’s and/or master’s) in Exercise and Sport Science and with expertise in pregnancy and postpartum exercises, as described for the Pregnancy and Postpartum Exercise Specialist by EuropeActive [68]. Those exercise physiologists represent the key providers of structured training for the pregnant participants, with the knowledge and skills to report and refer to healthcare providers during the course of the pregnancy, if necessary.

Item 2.3—Description of whether exercises are performed individually or in a group: The prenatal exercise program consists of a land-based program to be performed in a group. Each session is delivered to groups of 6–12 pregnant women. It is possible to adapt the structure of the group sessions to individual or online sessions, if needed or desired by the participants (e.g., in the case of another COVID-19 pandemic, participants can be connected online as a group).

Item 2.4—Description of whether exercises are supervised or unsupervised and how they are delivered: The exercise program is delivered face to face by an exercise professional whose qualifications were described in Section 2.2. Supervision is required to track the performance and adherence, to provide feedback on proper technique, to adapt the exercises if necessary, to ensure safety, and to refer to a healthcare professional if necessary. Supervision is an important competence of the prenatal exercise specialist that should prescribe and conduct exercise programs for pregnant women [69]. Moreover, the loss to follow-up was higher and adherence to intervention was lower for unsupervised versus supervised studies [23]. In parallel to the group program, the participants were advised to be physically active every day by means of walking, swimming, or cycling or by following YouTube videos at home [53,70].

Item 2.5—How adherence to exercise is measured and reported: The exercise professional checks the participation in each of the three available exercise sessions per week. The attendance rate to the prenatal exercise program is calculated by dividing the number of sessions attended by the number of sessions scheduled. Participants are encouraged to report the supervised and non-supervised sessions in an agenda (available for free download) [71] or in a mobile application (such as Strava). As an example, the number of steps per day and the type of sessions per week should be reported.

Item 2.6—Motivation strategies: Motivation strategies are applied through direct interaction between the exercise professional and the pregnant participants. Extensive counselling and knowledge transfer on the importance of regular and consistent exercise is provided through clear instructions. Pregnant participants are instructed to wear a pedometer or a step count mobile phone application and to register their daily step number. In each session, they are asked “Did you wear a pedometer all day today, except for showering, swimming, or sleeping?” and “Did you register the number of steps?”. Pregnant participants are also instructed to fill out the following questionnaires each month in order to check their progress with regards to their goals and potential contraindications for exercising:The PAR-Q+ Physical Activity Readiness Questionnaire for Everyone [72]The GET ACTIVE QUESTIONNAIRE FOR PREGNANCY by CSEP [73] that replaced the Physical Activity Readiness Medical Examination (PARmed-X) for Pregnancy after the publication of the Canadian guidelines [38]. This 2-page guideline for health screening facilitates communication between healthcare professionals, exercise specialists, and pregnant women. It includes guidance on exercise prescriptions, a healthy lifestyle during pregnancy, and exercise safety. The questionnaire is available at: https://csep.ca/wp-content/uploads/2021/05/GAQ_P_English.pdf (accessed on 15 March 2022).The CSEP also made available the HEALTH CARE PROVIDER CONSULTATION FORM FOR PRENATAL PHYSICAL ACTIVITY [74]The PPAQ Pregnancy Physical Activity Questionnaire developed by Chasan-Taber et al. in 2004 [75] is a widely used tool for the assessment and measurement of the physical activity levels amongst pregnant women, available at: https://journals.lww.com/acsm-msse/Fulltext/2004/10000/Development_and_Validation_of_a_Pregnancy_Physical.14.aspx (accessed on 15 March 2022).

These questions were provided in the “Active Pregnancy Agenda”, available for free online by Santos-Rocha et al. [71]. Moreover, pregnant women are instructed to track their Resting Heart Rate (RHR) and Training Heart Rate (THR) each week and are informed on the energy expenditure of weekly exercise sessions (and physical activity), as well as their fitness level. They are also referred to a nutritionist in order to follow a tailored nutrition plan. Participants are also recommended to perform basic fitness assessments (e.g., modified sit and reach, modified arm flexion extension, and the 1-mile Rockport walking test) to assess their progress, in accordance with the procedures described elsewhere [76,77].

Another motivational strategy is the creation of social network (WhatsApp and Instagram) private groups to provide alerts to allow for the sharing of information and social interaction. Moreover, short “webinars” are published in the Active Pregnancy YouTube channel [53], and the “Active Pregnancy Guide”, available for free online by Santos-Rocha et al. [78], is delivered to all women. The aims of these publications are to provide counselling, education, and motivation for exercising. The motivational strategies to increase adherence when delivering exercise sessions to pregnant women are addressed in the workshop for training exercise professionals [50].

Item 2.7a—Decision rule(s) for determining exercise progression: The exercise program is organized into three basic sessions of 60 min regarding three periods: adaptation (2 weeks), improvement (6 weeks), and maintenance (4 weeks). Each session follows a conventional structure [76] and addresses the health-related fitness components. The prenatal exercise was planned for each trimester of pregnancy according to the recommendations for physical activity by CSEP [38] and ACOG [41] and as described in our textbook [47,77,79].

Each 12 weeks of training was periodized into 4 weeks regarding the intensity using the Heart Rate Reserve (HRR) and the Borg scale [76]. In brief, the first trimester at 40–50% HRR; second trimester, from 40–50% HRR (fourth month) to 50–60% HRR (fifth and sixth months); and third trimester, from 50–60% HRR (seventh month) to 40–50% HRR (last two months of gestation). Previously sedentary women should start at 30% HRR and progress accordingly. Previously active women and athletes should continue the training, starting the prenatal program at 50–60% HRR, and progress in accordance with their fitness level and objectives [80]. If the HRR is not available, the goal is to reach a light-to-moderate training level Borg rating of perceived exertion (RPE) (11–13 out of 20) that corresponds to 50% of the estimated maximum oxygen uptake and 60% of the estimated Maximum Heart Rate (HRmax) using Gellish et al.’s equation [81] to an exertion level of RPE (14–15 out of 20) that corresponds to 75% of the estimated maximum oxygen uptake and 80% of the HRmax.

Regarding resistance training, the exercise physiologist adjusts the exercise intensity (load) as determined by the participant’s ability to complete two to four sets of eight to twelve repetitions for a given exercise (0–60% of one repetition maximum for lower body exercises). If the exercise results in pain, discomfort, or fatigue, the intensity should be lowered. In addition, participants receive feedback on their exercise progression.

A full description of exercise selection, adaptation, and progression is described elsewhere [52,79].

Item 2.7b—How the exercise program progresses: The intensity progress is monitored based on the HR, Borg scale, and the “talk test” [76,77]. Progression is achieved by increasing the intensity of the aerobic exercises, moving from simple to advanced/complex exercises with regards to task specificity, by increasing the number of sets and loads of the resistance exercise and the duration of the stretching exercise as tolerated.

Item 2.8—Description of each exercise to enable replication: The prenatal exercise program is described in a digital manual (e-book) by Santos-Rocha et al. [52] freely available on the internet, as well as in the Active Pregnancy YouTube channel [53]. Details on the organization of the sessions and the description of each exercise are provided regarding equipment, position, adaptations, technique, number of repetitions and sets, and safety considerations. Each exercise is illustrated by a picture of each phase or form of performance to allow for replication. The exercise testing and prescription process underlying the program is described elsewhere in a digital manual (e-book) by Santos-Rocha [82] freely available on the internet by Santos-Rocha et al. [77].

Item 2.9—Description of any home program component: No formal home-based component is included. Yet, the pregnant participants are advised to walk daily, at least 30 min, and to perform pelvic floor muscles training when not attending the supervised program. Other self-performed daily physical activities may include cycling, swimming, or gardening. Step counters are advised to estimate their physical activity level. However, due to the COVID-19 pandemic since 2020, and in accordance with the national and international ACSM Exercise is Medicine recommendations [83] that advise pregnant women to avoid indoor gyms and recommend limiting social gatherings, a series of exercise proposals to be active at home were published on our above-mentioned YouTube channels [53,70]. Moreover, since the COVID-19 pandemic caused fitness facilities to close and restructure services and certainly made an impact on the “Worldwide Survey of Fitness Trends” for 2021 and 2022 [84,85], the first trends became wearable technology and online training (i.e., developed for the at-home exercise experience, this trend uses digital streaming technology to deliver group, individual, or instructional exercise programs online). Thus, the exercise program can be delivered online if necessary, as long as exercise professionals are able to educate and supervise pregnant women in a remote setting [86].

Item 2.10—Description of whether there are any non-exercise components: The non-exercise component of the program refers to advice on the benefits of a healthy lifestyle (including nutrition, sleep, etc.), physical activity in general and prenatal exercise in particular, for the mother and the baby; how to measure the RHR and to determine the THR zones; and other topics by means of short lectures and using the above-mentioned materials:The available documents with recommendations and guidelines for physical activity during pregnancy, since they may have a positive impact [87].Active Pregnancy Guide [78], which includes educational information about the benefits of exercise, body adaptations, health and fitness, advice on the absolute and relative contraindications for exercise, etc.Active Pregnancy Agenda [71] includes pre-exercise assessments and fitness and exercise registries by means of questionnaires and reports.Active Pregnancy YouTube channel webinars, which include educational information [53].

Moreover, pregnant participants are advised to engage in preparation for a birth program during the third trimester provided by a qualified midwife [88]. Pregnant participants are also advised to engage in an early postpartum recovery exercise program as soon as possible after birth provided by a qualified exercise physiologist [89].

Item 2.11—Type and number of adverse events that occur during exercise: No adverse events occurred during the pilot intervention.

Item 2.12—Setting in which the exercises are performed: The land-based prenatal exercise program is supposed to be delivered in a fitness club setting with proper facilities, regarding safety and hygiene conditions [90], and equipment to groups of up to 15 pregnant women. It should include at least one group training room. The program is performed facing a mirror (optional) with basic equipment. The environment must be clean, ventilated, and comfortable regarding temperature, acoustic, and safety conditions (e.g., the ground floor). The floor conditions should be not slippery and be biomechanically appropriate for impact absorption. The exercise professional must keep visual contact with the group during the entire session and provide individual feedback. Workout music is optional. Music can also be used for motivation according to the preferences of the participants.

Some exercises can also be performed outdoors if the weather conditions are favorable. If participants prefer to exercise at home or outdoors, precautions should be taken regarding the type of floor. It is important to know the conditions of the weather and ground (not slippery floor) in advance and avoid bad weather (too cold or too hot), noise, and air pollution.

Item 2.13—Detailed description of the exercise intervention: The structure of the prenatal exercise program and its periodization was defined, and a portfolio of the exercises was described for each part of a typical 60-min session, as follows:(1)warm-up: 5–10 min (8–17%);(2)aerobic training p: 25 min (42%);(3)neuromotor training (posture, balance, and coordination): 5 min (8%);(4)resistance training (core, lower, and upper limbs): 10 min (17%);(5)pelvic floor training: 5 min (8%);(6)stretching: 5 min (8%);(7)breathing and relaxation: 5 min (8%).

For each exercise, pictures; a description of the objectives; the main critical aspects (technique and safety); the equipment; the options of different positions (standing, seating, and laying); and the options for increasing or decreasing the intensity and complexity regarding each trimester of pregnancy are provided [52,53].

Item 2.14a—Description of whether the exercises are generic or tailored: The core exercise is generically described, and variations are provided to tailor the exercise to the anatomical adaptations of the pregnant participant.

Item 2.14b—Detailed description of how exercises are tailored to the individual: Each exercise includes variations to increase or decrease the intensity or complexity regarding each trimester of pregnancy, their skills, and their fitness level.

Item 2.15—Decision rule for determining the starting level: The starting level is defined by the previous experience and fitness level of the pregnant participant. The exercise physiologist should follow the pre-exercise assessment protocol described by ACSM [76] and in [77].

Item 2.16a—Description of how the adherence or fidelity is assessed/measured: Described in Item 2.5.

Item 2.16b—Description of the extent to which the intervention was delivered as planned: The pilot prenatal exercise program was delivered as planned and as feasible during the pandemic thanks to an “online home-based” option during the times of physical confinement. Most participants attended more than 90% of the sessions. There was no drop-out.

Item 3—Illustration of any intended interactions between different components.

The above-mentioned educational tools (manuals and YouTube channels) and, also, a guide for health professionals on physical activity promotion during pregnancy [91] were produced by eleven exercise specialists and ten health professionals (nutritionists, psychologists, gynecologists, dentists, midwives, and physiotherapists) that contributed as authors of the publication. Moreover, before publishing, feedback was provided by the pregnant women, other exercise and health professionals, and exercise sciences students. The exercise program also includes a 6-h workshop to sensitize exercise professionals about the structure of each session and the best use of the exercise program guide. This workshop intends to sensitize health professionals about the promotion of physical activity during pregnancy [50].

Item 4—Description and consideration of the context’s characteristics in intervention modeling.

This content was addressed in Items 2—2.12.

### 3.2. Second Stage: Feasibility and Piloting

Item 5—Description of the pilot test and its impact on the definite intervention.

The pilot tests aimed to determine the feasibility and acceptability of the exercise program, and the supporting materials were tested by three groups of participants (Figure 1).

The first version of the exercise program was developed by two exercise specialists, and it was delivered by means of a slideshow presentation among the research team. The first version of the exercise program in the format of a digital manual (e-book) was shared among six exercise specialists and four health/nursing specialists. Moreover, a 75-min workshop was delivered to a group of exercise professionals in order to collect their feedback on the exercise program.

Four-hour workshops were delivered to four groups of students in exercise sciences (bachelor’s and master’s programs of the Polytechnic Institutes of Santarém, Leiria and Beja) and one group of midwives (master’s program of the Polytechnic Institute of Santarém). After each workshop, an anonymous questionnaire was provided to each professional or student in order to collect feedback on the exercise program regarding the structure, duration, frequency, type of exercises, variations, progression, equipment, feasibility, tutorials, etc. [50]. The participants were invited to provide feedback in 12 questions using a 5-point Likert scale, ranging from “absolutely agree” to “absolutely disagree”, except in questions 11 and 12, which included three options (i.e., “both”, “app”, “e-book” and “both”, “printed book”, “poster”, respectively). The questionnaire also included an open question (13) and three questions about the occupation, gender, and age of the respondents.

The majority of the 96 participants in the workshops agreed or absolutely agreed with the following aspects:(1)the exercise program is well-structured (3% and 97%, respectively);(2)the duration (60 min sessions) of the exercise program is adequate (10% and 90%, respectively);(3)the frequency (3 times per week) of the exercise program is adequate (3% and 97%, respectively);(4)the variety of exercises provided in the program are adequate for the targeted population (3% and 97%, respectively);(5)the variations of the exercises provided in the program are adequate for the targeted population (3% and 97%, respectively);(6)the equipment is adequate for the targeted population (0% and 100%, respectively);(7)the structure of each session model is adequate for the targeted population (0% and 100%, respectively);(8)the structure of each session model is clear and easy to follow by the professionals (7% and 93%, respectively);(9)the explanations about each exercise are clear to the professionals (3% and 97%, respectively);(10)the exercise program has the potential to be replicated (10% and 90%, respectively);(11)the exercise program should be available to professionals by means of an e-book and an informatics application (0% and 100%, respectively);(12)the exercise program should be available to professionals by means of a printed book and posters (90% and 10%, respectively).

Finally, the open question “13—Would you like to provide any other comments?” was answered by comments such as: “add more practical videos”, “include basic exercises to be performed every day and at home”, and “printed format and an informatics application (under construction) should be made available”.

In general, the feedback was positive and encouraged the integration of tutorial videos in the manual. After publishing the manual, the YouTube channel was launched in October 2020 (Figure 2). Further exercises and variations were subsequently integrated and tutorial videos added.

The pilot exercise intervention of 12 weeks is described in the Methods section.

The “Active Pregnancy Guide” was delivered to all the participants [78]. During the sessions, no incidents or injuries were reported.

The report on the pilot intervention experience of participating in an exercise program during pregnancy, including satisfaction, adherence, and adverse events, was performed in accordance with Haakstad et al. [49]. After the intervention, the participants were asked to provide feedback regarding their satisfaction with the training sessions, types of exercise, and perception of physical fitness improvement as follows:72% of the participants reported being very satisfied with the program, whereas 28% reported being satisfied83% reported being very satisfied with the exercise professional, whereas 17% reported being satisfied83% reported higher motivation in a group setting, whereas 17% reported no change compared to personal training95% reported an improvement in their physical fitness, notably in terms of strength and control of their body weight, whereas 5% reported no change78% of the women reported increased levels of physical activity during pregnancy, whereas 22% reported no changes90% reported that they felt more energy for daily activities and less stress, whereas 10% reported no changes

All participants reported that they would recommend the training program and confirmed their participation during any future pregnancy.

### 3.3. Third Stage: Evaluation

Item 6—Description of the control condition (comparator) and reasons for the selection.

Recent systematic reviews have been showing the effectiveness of physical activity in several maternal health outcomes and in a few fitness parameters [11,12,13,14,24,25]. However, it is not clear which features are included in the most effective exercise programs regarding outcomes and long-term adherence. A protocol of a RCT study testing the effectiveness of the program is described elsewhere.

Item 7—Description of the strategy for delivering the intervention within the study context.

The healthy lifestyle counselling program was planned to be delivered in a healthcare environment, as well as online. The healthy lifestyle counselling program should be delivered by midwives and exercise physiologists/exercise specialists with experience in prenatal exercise. The exercise program was planned to be delivered in a fitness club environment, hospital, or online. This program should be included in the regular schedule of a gym, taking into consideration that most women keep up any occupational activity during pregnancy. The exercise program should be delivered by exercise physiologists supported by the materials produced by the research team (training, e-book, and videos) when planning all sessions. In the case of a pandemic, the exercise program can be delivered online to all groups.

Item 8—Description of all materials or tools used in the delivery of the intervention.

This content was address in Items 2.1 and 2.12.

Item 9—Description of fidelity of the delivery process compared the study protocol.

The aim of the present study was to build and validate an exercise program. The exercise program describes each exercise and its variations. It also includes a plan organized into three different periods, including the session structure and the suggested exercises for each stage, while keeping in mind the predefined objectives. Thus, it works as a predesigned intervention to be followed by the exercise professional. The exercise program aims to be safe and effective regarding several health and fitness parameters and should be tested in multicenter trials. The protocol of the study will be described elsewhere.

Item 10—Description of a process evaluation and its underlying theoretical basis.

The process of evaluation was planned to determine the effectiveness of the exercise program. The outcome measures will be assessed at the baseline (starting during the first or second trimester of gestation) and after 12 weeks of intervention (during the second or third trimester of gestation, respectively).

The main outcome measures to evaluate the effectiveness of the exercise program are: health perception, pregnancy-related symptoms and contraindications for exercising (questionnaires), the RHR and THR (using a monitor), fatigue, physical activity volume, happiness, stress and anxiety, low back pain, motivation, satisfaction with the program and instructor, quality of life (using questionnaires), weight gain (using a scale), cardiorespiratory fitness, hand grip strength, upper and lower limb strength, flexibility (using validated test batteries and a global questionnaire), posture, balance, and the biomechanical parameters of gait (using a plantar pressure assessment device).

Item 11—Description of internal facilitators and barriers potentially influencing the delivery of the intervention as revealed by the process evaluation.

Facilitators potentially influencing the delivery of the intervention are related to the availability of facilities and equipment required to deliver each mode of the exercise program, to the strategy of the fitness club in offering at least one mode of the exercise program, and also related to the effectiveness of the communication process of each setting where the intervention takes place. The exercise program can be delivered in a hospital setting or in a remote setting. Other facilitators are related to the competence of health professionals such as gynecologists, general practitioners, midwifes, physiotherapists, nutritionists, and psychologists in supporting women who take an active role via shared decision-making in the management of an active lifestyle during and after pregnancy [92,93,94]. Social support, access to resources, information, proper diet, scheduling, and the weather were identified as powerful facilitators [95,96]. Promoting the guidelines and educational materials providing information about physical activity during pregnancy is expected to help pregnant women to engage in the exercise program [87]. A lack of support from health professionals and from family, physical limitations and restrictions, lack of resources, lack of energy, and lack of time are identified exercise barriers potentially influencing the delivery of the intervention.

Item 12—Description of external conditions or factors occurring during the study that might have influenced the delivery of the intervention or mode of action.

External conditions or factors occurring during the study that might influence the delivery of the intervention are related to the fact that the recruitment and referral processes also depend on the counselling provided by healthcare professionals and on the pregnant women’s motivation.

Item 13—Description of costs or required resources for the delivery of the intervention.

The exercise program was designed to be delivered in a fitness club with an exercise room. These sport facilities and equipment must follow all required safety and hygiene rules and legislation for maintenance and exploitation, which implies costs. The exercise program was planned and structured to be delivered by qualified exercise professionals with knowledge of exercise sciences and prenatal exercise and with practical experience with pregnant women. There are also the costs of the research equipment to be used, such as hand grip and gait pattern assessment device and software, among others. The dissemination of knowledge and outcomes via educational materials, papers, and congresses also requires funding. Thus, the implementation of the exercise program requires specific costs due to facilities, equipment, dissemination, and qualified professionals.

## 4. Discussion

It is of major importance to develop and validate reproducible and effective physical exercise programs aimed at promoting health and fitness during pregnancy. A physical exercise program can be considered a complex intervention, since it is tailored to a specific population and setting, and it is affected by several components regarding effectiveness and safety. Thus, the need to develop and validate well-defined and replicable exercise protocols emerges in order to bridge the identified gaps.

The CReDECI2 process has the potential to help practitioners in developing and planning complex interventions, such as an exercise program, as well as researchers in planning exercise trials. Its exercise components and other tools can be adjusted to the context and to the characteristics of the target population. However, it allows testing the feasibility and acceptance of the intervention and not its effectiveness.

The protocol of a RCT study should be developed to test the effectiveness of the program on promoting health and fitness during pregnancy. We hypothesized that a high-frequency, moderate-intensity, supervised multicomponent exercise intervention combined with a healthy lifestyle counselling program would result in an improved maternal physical activity volume and fitness and health parameters (including mental, physical, and social parameters) when compared with a low-frequency, light-intensity supervised multicomponent exercise intervention. We also hypothesized that an online intervention would be as effective as in person. Moreover, we hypothesized that supervised exercise interventions would be more effective than unsupervised exercise. This protocol will describe the inclusion and exclusion criteria and the recruitment process to engage pregnant women in the Active Pregnancy exercise program to be delivered in multicenter intervention groups, as well as online. A healthy lifestyle counselling program that includes physical activity will be delivered online to all groups.

A non-exercise control group either receiving “standard care” or to be included on a “waiting list” will not be enrolled in the study, as of a typical RCT study, due to ethical reasons. Namely, there is plenty of evidence supporting the benefits of physical activity during pregnancy. It is health and exercise professionals’ responsibility to advise and encourage women towards physical activity. The standard care, unfortunately, does not focus on physical activity promotion. On the other hand, it does not make sense to tell a pregnant woman that she is on a waiting list for 12 weeks, since she might no longer be pregnant.

It is worth noting that the presented exercise program proposal may be modified, depending on the specific needs of pregnant women. Certainly, it would be worthwhile in the future to develop programs aimed at implementing exercises for women with a complicated course of pregnancy, e.g., gestational diabetes mellitus, low back pain, dyslipidemia, gestational hypertension, or perinatal depression. An interesting task also seems to be the development of similar protocols based on high-intensity interval training, which is gaining popularity in the population of pregnant women [97].

Moreover, future developments foresee a website and an informatics application (“app”) to provide guidance to women and professionals.

## 5. Limitations

The main limitation of the study is that, although the revised guideline CReDECI2 was followed to validate a physical exercise program aimed at improving the health and fitness parameters of pregnant women, this process does not guarantee the effectiveness of the intervention. Moreover, the process does not guarantee the absence of obstacles in the design, implementation, or evaluation of a future larger-scale study. It should be noted that, during the process of the development of the physical exercise program, the hypothesized effects of the intervention were not assessed regarding any of the outcomes of interest highlighted in the literature (i.e., quality of life, gestational diabetes, gait pattern and prevention of falls, depressive symptoms, and health-related and skill-related fitness parameters). However, it has been found that, on a consensus basis, experts and potential program users see this type of intervention as relevant and necessary in a healthcare context. This innovative approach of using the CREDECI2 framework to develop an exercise program has made the intervention more likely to be acceptable and deepens the understanding of how such a complex intervention performs during primary care; however, the effectiveness of this exercise program in improving maternal health and the fitness parameters needs to be evaluated in a RCT. If positive effects are observed, the Active Pregnancy program can be considered for incorporation into the healthcare system. Although this development process enabled us to make a number of improvements to our intervention and to achieve a design that is likely to be accepted in the setting in which it is to be delivered and tested, avoiding duplicating efforts for subsequent research, the process has some limitations. It was time-consuming, i.e., the process described took 12 months to complete. The pilot intervention also included a small number of participants (i.e., less than 30), which may have limited the generalizability of the conclusions.

## 6. Future Research

Physical activity and exercise interventions with pregnant women lack standardized methodologies for the development, delivery, and assessment of such programs due to their complexity [3,4,24,25]. The CERT is a template that can be used when constructing and reporting such exercise programs [33,34]. The revised guideline CReDECI2 can be used to validate prenatal exercise interventions.

This study included a pilot intervention to test the feasibility and safety of a multicomponent prenatal exercise program. The study protocol of a RCT testing its effectiveness should be further described.

Recent publications have been showing the effectiveness of physical activity in several maternal health outcomes and in cardiorespiratory parameters [11,12,13,14,24,25]. However, it is not clear which features are included in the most effective exercise programs regarding the outcomes and long-term adherence. On the other hand, there is a lack of RCT testing the effectiveness of prenatal exercise programs on other health-related and skill-related fitness components, namely muscular fitness, gait, balance, and coordination. There is also a lack of RCT studies testing the effectiveness of prenatal multicomponent exercise programs with different features regarding volume (high-frequency, moderate-intensity versus low-frequency, light-intensity), supervision (structured supervised exercise interventions versus unsupervised exercise), and context of the intervention (online versus in-person interventions). In particular, there are no studies so far testing the effectiveness of online interventions that were substantially increased after the COVID-19 pandemic in 2020. Thus, the study protocol should address these hypotheses.

The Canadian and the ACOG guidelines state that some pregnancy-related symptoms and complications may not justify stopping the practice of exercise, as these pregnant women will benefit from being physically active. However, there is also a lack of studies addressing the effectiveness, adherence, and safety of prenatal exercise programs on these special populations, namely pregnant women with gestational diabetes, anxiety, depression, overweightness, and disabilities.

Finally, future RCT studies should compare groups of pregnant women receiving exercise interventions with different features, and there should not be comparisons with non-exercise control groups either receiving “standard care” or included on a “waiting list” due to ethical reasons.

## 7. Conclusions

The CREDECI2 framework for complex interventions provides a structured approach to guide the development of novel interventions for public health. It was extensively followed to validate a prenatal exercise program. The results of the present work can be useful in assisting exercise and health professionals and researchers in planning, promoting, and implementing complex interventions and trials, such as a prenatal exercise program. The implications of its practice and effectiveness on maternal health and the fitness parameters will emerge when the results of our randomized controlled trials are published.

## Figures and Tables

**Figure 1 ijerph-19-04902-f001:**
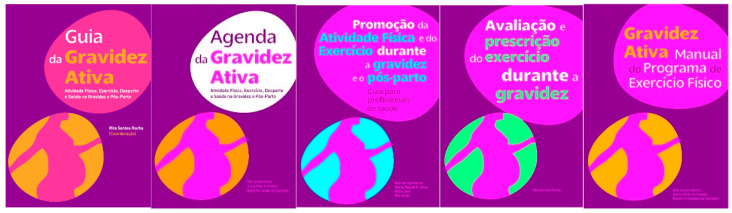
Layout of the educational materials produced with the support of IPDJ—Instituto Português do Desporto e da Juventude (Portuguese Institute of Sport and Youth).

**Figure 2 ijerph-19-04902-f002:**
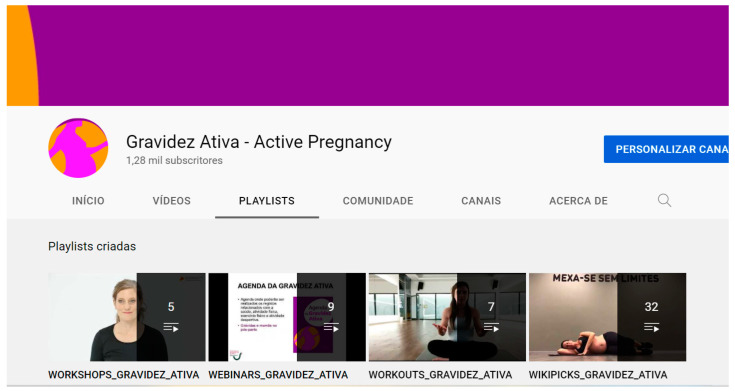
Layout of the Active Pregnancy YouTube channel.

## Data Availability

Not applicable.

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
