# Peer review of "Active Pregnancy: A Physical Exercise Program Promoting Fitness and Health during Pregnancy—Development and Validation of a Complex Intervention"

_ijerph, 2022, doi:10.3390/ijerph19084902_

Round 1

Reviewer 1 Report

Dear authors, of course, the topic you have chosen is relevant, interesting and deserves deep development.

In my opinion, the publication, from a scientific point of view, would be strengthened by objective data on the condition of pregnant women participating in the study (functional, mental and other), you have presented only subjective survey results (although we all understand the instability of the mood of any pregnant woman, even performing physical exercises :)).

 I also have a question about the logic of a set of physical exercises: how physiologically justified is it to perform a block of aerobic exercises before neuromotor and resistance exercises? (although, of course, this is your copyright).

 I repeat, in general, the article is interesting and relevant.

Author Response

Author's Reply to the Review Report (Reviewer 1)

Dear authors, of course, the topic you have chosen is relevant, interesting and deserves deep development.

  • Thank you very much for the comment.

In my opinion, the publication, from a scientific point of view, would be strengthened by objective data on the condition of pregnant women participating in the study (functional, mental and other), you have presented only subjective survey results (although we all understand the instability of the mood of any pregnant woman, even performing physical exercises :)).

  • More data regarding the characteristics of the participants in the pilot intervention were included in the methods section. Line 193

I also have a question about the logic of a set of physical exercises: how physiologically justified is it to perform a block of aerobic exercises before neuromotor and resistance exercises? (although, of course, this is your copyright).

  • The exercise intervention was planned in accordance to practice of many years of experience. Nevertheless, the components of an exercise training session are explained in ACSM´s Exercise Testing and Prescription Guidelines, a fundamental textbook supporting exercise physiologists and researchers, and developed in our textbook published by Springer.

I repeat, in general, the article is interesting and relevant.

  • Thank you very much for your time and feedback.
  • We hope that the revisions made in the paper will help to improve it.

Reviewer 2 Report

General comments

The manuscript is, in a way, a series of citations of other works, constantly referring the reader to the given references and not providing specific guidelines, tips, and summaries. It comes down to searching the literature and browsing the content of the works in the links, which is very tiring, makes it very difficult to understand the scope and message of the manuscript, and discourages continuing to read the manuscript. It should be corrected so that the work becomes clear and brings positive values to the current state of knowledge. The wording "described elsewhere" comes up far too often. It isn't easy to read this work and understand what part of your research is and what has already been mentioned in the literature.

Specific comments:

88-90; "…and 29 pregnant women participating in the pilot intervention." Please specify the age of the patients, the stage of pregnancy in weeks, and which, in turn, was the participants' pregnancy and what their physical activity looked like before becoming pregnant. Were the participants physically active in any possible early pregnancies?

133-152: It is a set of guidelines and not the results of research work. Therefore, I suggest moving this fragment to the Materials and Methods section to facilitate the reading of the manuscript content and clearly indicate the work's results, no international guidelines on this matter.

353-362: This is an example of an actual recommendation in the ACTIVE PREGNANCY program, and this is the way the full results should be presented, which is to show the essence of this program, not only a part of it, and the rest in the guidelines in the literature references that need to be found and followed. The work would then gain coherence and readability, which is far from the desired minimum in this version.

443-450: Should be moved to the Materials and Methods section.

479-490: Should be moved to the Discussion section.

498-508: Should be moved and the end of the Discussion section. These findings are not the results but further plans of distribution of guidance.

512-519: As mentioned above.

533-550: As written regarding 498-508 lines.

571: Discussion: A very perfunctory and superficial discussion, not including the essence of the problem, such as the advantages and disadvantages of the developed program, implementation problems on the market, current preparation of the staff for its application, who can use it, what skills and experience should have when the program may be a threat to a woman using it. Please look at this problem from the recipient's perspective, not the developer or implementer.

595-604: Suggest moving to do the Discussion section.

618-626: This passage should be kept in the context of what the authors of the manuscript plan to research further, not what should be done and by whom in the world unless the authors plan to continue their research.

Author Response

Author's Reply to the Review Report (Reviewer 2)

General comments

The manuscript is, in a way, a series of citations of other works, constantly referring the reader to the given references and not providing specific guidelines, tips, and summaries. It comes down to searching the literature and browsing the content of the works in the links, which is very tiring, makes it very difficult to understand the scope and message of the manuscript, and discourages continuing to read the manuscript. It should be corrected so that the work becomes clear and brings positive values to the current state of knowledge. The wording "described elsewhere" comes up far too often. It isn't easy to read this work and understand what part of your research is and what has already been mentioned in the literature.

  • Thank you very much for the comment, although it is very disappointing to read that someone did not like or understand a work where we put a lot of effort and enthusiasm and reflects many years of practice and research.
  • We hope that the revisions made in the paper will help to improve these aspects.
  • On the other hand, we will have to maintain the reference "described elsewhere", since it is impossible to describe everything in the same paper.

Specific comments:

88-90; "…and 29 pregnant women participating in the pilot intervention." Please specify the age of the patients, the stage of pregnancy in weeks, and which, in turn, was the participants' pregnancy and what their physical activity looked like before becoming pregnant. Were the participants physically active in any possible early pregnancies?

  • More data regarding the characteristics of the participants in the pilot intervention were included in the methods section. Line 193

133-152: It is a set of guidelines and not the results of research work. Therefore, I suggest moving this fragment to the Materials and Methods section to facilitate the reading of the manuscript content and clearly indicate the work's results, no international guidelines on this matter.

  • The aim of this qualitative study was to build and validate a multicomponent prenatal exercise program to promoting health and fitness during pregnancy, and for this, we exhaustively followed the CReDECI2 process. Item 1 refers to the description of the intervention’s underlying theoretical basis. There are official guidelines on physical activity during pregnancy published by national and international obstetrics, gynecology, or sports medicine institutions that were recently updated. Thus, they must be inserted in this section.

353-362: This is an example of an actual recommendation in the ACTIVE PREGNANCY program, and this is the way the full results should be presented, which is to show the essence of this program, not only a part of it, and the rest in the guidelines in the literature references that need to be found and followed. The work would then gain coherence and readability, which is far from the desired minimum in this version.

  • We hope that the revisions made in the paper will help to improve these aspects.

443-450: Should be moved to the Materials and Methods section.

  • Again, the aim of this qualitative study was to build and validate a multicomponent prenatal exercise program to promoting health and fitness during pregnancy, and for this, we exhaustively followed the CReDECI2 process. Item 5 refers to the description of the pilot test and its impact on the definite intervention and it is included in the “Second stage: Feasibility and piloting”. Nevertheless, based on your feedback, we have improved the methods section. Line 193

479-490: Should be moved to the Discussion section.

  • These paragraphs were moved to the discussion section. Thank you for the suggestion.

498-508: Should be moved and the end of the Discussion section. These findings are not the results but further plans of distribution of guidance.

  • These paragraphs follow item 8, 9, and 10 of CREDECI2

512-519: As mentioned above.

  • These paragraphs follow items 8, 9, and 10 of CREDECI2

533-550: As written regarding 498-508 lines.

  • These paragraphs follow items 8, 9, and 10 of CREDECI2

571: Discussion: A very perfunctory and superficial discussion, not including the essence of the problem, such as the advantages and disadvantages of the developed program, implementation problems on the market, current preparation of the staff for its application, who can use it, what skills and experience should have when the program may be a threat to a woman using it. Please look at this problem from the recipient's perspective, not the developer or implementer.

  • The discussion section was improved as suggested. Indeed, the “advantages and disadvantages of the developed program, implementation problems on the market, current preparation of the staff for its application, who can use it, what skills and experience should have when the program may be a threat to a woman using it” are very important issues with which the authors are pretty much aware of, since they have been working in practice with pregnant woman and not just as researchers in the field. However, these issues were addressed in other publications and were not the objectives of the present study.

595-604: Suggest moving to do the Discussion section.

  • These paragraphs follow item 12 of CREDECI2

618-626: This passage should be kept in the context of what the authors of the manuscript plan to research further, not what should be done and by whom in the world unless the authors plan to continue their research.

  • These paragraphs try to explain further research lines of the research team, in line with what we have been publishing in projects, webinars, conferences, papers, textbooks, educational materials, videos, master and doctoral theses, etc., i.e., published elsewhere…
  • Thank you very much for your time and feedback.
  • We hope that the revisions made in the paper will help to improve it.

Reviewer 3 Report

Page 5, line 208 to 212: suggest to add these to the introduction as it sheds light on why it was important to implement and evaluate a physical exercise program in pregnant women (include the reference, since it is a very recent publication which further emphasizes that there was a need for this program)

Page 10, lines 457 - 468: Suggest to present this feedback on satisfaction survey in a tabular format for better visual presentation and understanding

Page 10, Item 6, lines 479 - 486: Suggest to include all study hypothesis including those for 3rd stage of evaluation in the last paragraph of the introduction section of the paper

Author Response

Author's Reply to the Review Report (Reviewer 3)

Page 5, line 208 to 212: suggest to add these to the introduction as it sheds light on why it was important to implement and evaluate a physical exercise program in pregnant women (include the reference, since it is a very recent publication which further emphasizes that there was a need for this program)

  • Thank you very much for the suggestion. We moved this statement to the introduction section, as suggested. Line 52

Page 10, lines 457 - 468: Suggest to present this feedback on satisfaction survey in a tabular format for better visual presentation and understanding

  • Numbered bullets were inserted for better visual presentation.

Page 10, Item 6, lines 479 - 486: Suggest to include all study hypothesis including those for 3rd stage of evaluation in the last paragraph of the introduction section of the paper

  • We moved the hypothesis of a future RCT protocol, to the discussion and future research sections, as suggested by other reviewers, because this qualitative paper's objective is to validate the intervention to be tested in future RCT studies.
  • Thank you very much for your time and feedback.
  • We hope that the revisions made in the paper will help to improve it.

Reviewer 4 Report

IJERPH--1662501_review

Title: ACTIVE PREGNANCY: a physical exercise program promoting fitness and health during pregnancy. Development and validation of a complex intervention

Comments and Suggestions for Authors

Dear authors,

I have carefully read your paper that highlight the importance and need to encourage pregnant women to participate in exercise programs and promote prenatal physical activity.

The aim of your study was to build and validate a multicomponent prenatal exercise program to promoting health and fitness during pregnancy, for this, you exhaustively followed the CReDECI2 process.

In my opinion, although your study included a pilot intervention to test the feasibility and safety of the exercise program, the results shown are insufficient to validate this program. I consider that this manuscript shows us a protocol to develop and validate this exercise program, however, as your recognize, it is necessary to develop a study protocol and a pilot study to test the effectiveness of the program.

I agree with you that further researches are needed in this field to test the effectiveness of physical activity in the maternal health and in special populations as pregnant women with gestational diabetes, anxiety, depression, overweight, and disabilities, addressing the effectiveness, adherence, and safety in the prenatal exercise programs.  This could be useful to assist exercise and health professionals and researchers in planning, promoting, and implementing complex interventions with exercise programs.

In general, the manuscript is well-written. The text is understandable and organized, and it is easy to follow authors’ thoughts and reasoning. In my opinion, the introduction, and discussion sections are well-described. However, I found some issues in materials and methods, results and conclusions sections that should be addressed to improve the paper, in my opinion.

Specific comments:

Title:  

- Page 1, lines 2-4: In think the title could be more specific. I suggest “a protocol for development and validation of a complex intervention”

Introduction

  • The introduction section is well-structured and comprehensive.
  • Page 2, line 45: Use the full term the first time it appears in the text before using the acronym. In this line: randomized controlled trials (RCT’s). Please check it in the whole manuscript.                        

Materials and methods

  • Page 2, lines 83-89: I suggest you to add the information about the inclusion and exclusion criteria (which are in the results section, page 10, lines 445-447) in this section of the methodology. It would be more appropriate to add this information in the methods section.
  • Page 2, line 88. You mentioned in this line that: “29 pregnant women participating in the pilot intervention”, but in page 10, lines 448-449 you mentioned that 27 pregnant women participated in the pilot intervention. Please check it.
  • Page 2, lines 91-92: In page 2, lines 61-62 appears for the first time “revised guideline of Criteria for Reporting the Development and Evaluation 61 of Complex Interventions in healthcare (CReDECI 2)”. Once an acronym is used for a word, this acronym should be used in the rest of the manuscript. Please check it out as this is repeated several times throughout the text.
  • Page 3, line 100: 2.1.2. Participants. I think it is a grammar mistake, please check it.
  • Page 3, line 113: The videos of exercises in digital format are really nice, congratulations.
  • Page 3, lines 121-128: Was the informed consent obtained from all participants? If so, how was it recorded? Was the study conducted in accordance with the Helsinki declaration? If so, please add this information.

Results

  • Page 4, lines 196-197. You mentioned that the prenatal exercise program consists of a land-based program to be performed in group. However, due to the pandemic situation, this could be limited. Did you do the sessions online then, but did you all connect at the same time as a group? In my opinion, this information is central to your Please clarify it
  • Page 5, lines 241-242: The link is not available. Please, check it.

https://journals.lww.com/acsmmsse/Fulltext/2004/10000/Development_and_Validation_of_a_Pregnancy_Physical.14.aspx

  • Page 6, line 268 and 274: You mentioned the Borg scale and the Borg rating of perceived exertion (RPE). Please, add some references.
  • Page 9, lines 452-455: In the pilot intervention experience you collected information before and after about satisfaction, adherence, adverse events, types of exercise, and perception of physical fitness improvement. However, you have not collected important information about the baseline situation of the participants. As you mention on page 11, lines 520-532, data must be collected in this baseline situation to later establish a comparative pre-post analysis. In my opinion, you should have included, at least, some of these variables in this study and shown in your results, which would have more conclusively supported the validity of this protocol.
  • Page 11, lines 525-532: In this paragraph appear the main outcome measures to evaluate the effectiveness of the exercise program. Please report some references that support these outcomes, if possible.

Discussion

- Your discussion section is in general adequate and complete.

- You have not described the limitations of your study; please add them before the conclusions section.

Conclusions

Page 13, lines 629-633: I think conclusions should be rephrased in a more careful way. Your results do not support the statement that “The results of the present work will be useful to assist exercise and health professionals and researchers in…..” You mentioned that it is necessary to develop a study protocol and a pilot study to test the effectiveness of the program.  This should be considered. The results of the study suggested ……

I hope that my comments could help to improve the paper.

I would like to congratulate you for all the additional material and YouTube material prepared for this work.

Congratulations for your research.

Author Response

Author's Reply to the Review Report (Reviewer 4)

Comments and Suggestions for Authors

Dear authors,

I have carefully read your paper that highlight the importance and need to encourage pregnant women to participate in exercise programs and promote prenatal physical activity.

The aim of your study was to build and validate a multicomponent prenatal exercise program to promoting health and fitness during pregnancy, for this, you exhaustively followed the CReDECI2 process.

In my opinion, although your study included a pilot intervention to test the feasibility and safety of the exercise program, the results shown are insufficient to validate this program. I consider that this manuscript shows us a protocol to develop and validate this exercise program, however, as your recognize, it is necessary to develop a study protocol and a pilot study to test the effectiveness of the program.

I agree with you that further researches are needed in this field to test the effectiveness of physical activity in the maternal health and in special populations as pregnant women with gestational diabetes, anxiety, depression, overweight, and disabilities, addressing the effectiveness, adherence, and safety in the prenatal exercise programs.  This could be useful to assist exercise and health professionals and researchers in planning, promoting, and implementing complex interventions with exercise programs.

In general, the manuscript is well-written. The text is understandable and organized, and it is easy to follow authors’ thoughts and reasoning. In my opinion, the introduction, and discussion sections are well-described. However, I found some issues in materials and methods, results and conclusions sections that should be addressed to improve the paper, in my opinion.

  • Thank you very much for the comments. We hope we addressed your suggestions as best as possible.

Specific comments:

Title:  

- Page 1, lines 2-4: In think the title could be more specific. I suggest “a protocol for development and validation of a complex intervention”

  • In accordance with the authors, Comprehensive reporting of complex interventions enhances transparency and is essential for researchers and policy-makers (*).  “CReDECI 2 is a formally consented reporting guideline aiming to improve the reporting quality of the development and evaluation stages of complex interventions in healthcare. Since the guideline does not focus on a specific study design, design-specific reporting guidelines may additionally be used.” … “Characteristics that make interventions complex are different professions or different organizational levels targeted by the intervention (context of the intervention) and/or a need to tailor the intervention for specific settings (flexibility of the intervention)…. Most nonpharmacological, behavioral change and educational interventions are likely to be complex interventions …” (**).
  • (*) Möhler, R., Bartoszek, G. & Meyer, G. Quality of reporting of complex healthcare interventions and applicability of the CReDECI list - a survey of publications indexed in PubMed. BMC Med Res Methodol13, 125 (2013). https://doi.org/10.1186/1471-2288-13-125
  • (**) https://go.gale.com/ps/i.do?p=AONE&u=googlescholar&id=GALE|A541597738&v=2.1&it=r&sid=AONE&asid=cb9bde2c
  • Therefore, our title is in line with previous works that used this methodology, e.g.,
  • Byrne M, Cupples ME, Smith SM, Leathem C, Corrigan M, Byrne MC, Murphy AW. Development of a complex intervention for secondary prevention of coronary heart disease in primary care using the UK Medical Research Council framework. Am J Manag Care. 2006 May;12(5):261-6. PMID: 16686583.
  • Paul G, Smith SM, Whitford D, O'Kelly F, O'Dowd T. Development of a complex intervention to test the effectiveness of peer support in type 2 diabetes. BMC Health Serv Res. 2007 Aug 31;7:136. doi: 10.1186/1472-6963-7-136. PMID: 17764549; PMCID: PMC2080630.
  • Redfern J, Rudd AD, Wolfe CD, McKevitt C. Stop Stroke: development of an innovative intervention to improve risk factor management after stroke. Patient Educ Couns. 2008 Aug;72(2):201-9. doi: 10.1016/j.pec.2008.03.006. Epub 2008 Apr 28. PMID: 18440753.
  • Corrrigan M, Cupples ME, Smith SM, Byrne M, Leathem CS, Clerkin P, Murphy AW. The contribution of qualitative research in designing a complex intervention for secondary prevention of coronary heart disease in two different healthcare systems. BMC Health Serv Res. 2006 Jul 18;6:90. doi: 10.1186/1472-6963-6-90. PMID: 16848896; PMCID: PMC1543625.
  • Santos-Rocha R, Freitas J, Ramalho F, Pimenta N, Costa Couto F, Apóstolo J. Development and validation of a complex intervention: A physical exercise programme aimed at delaying the functional decline in frail older adults. Nurs Open. 2019 Sep 30;7(1):274-284. doi: 10.1002/nop2.388.

  • The CREDECI2 framework for complex interventions provides a structured approach to guide the development of novel interventions in public health, such as a prenatal exercise program. Implications for practice and effectiveness on maternal health and fitness parameters will emerge when the results of our randomized controlled trials are published.
  • This information was added to the conclusion section.

Introduction

The introduction section is well-structured and comprehensive.

Page 2, line 45: Use the full term the first time it appears in the text before using the acronym. In this line: randomized controlled trials (RCT’s). Please check it in the whole manuscript.       

  • Corrected. Line 45                 

Materials and methods

Page 2, lines 83-89: I suggest you to add the information about the inclusion and exclusion criteria (which are in the results section, page 10, lines 445-447) in this section of the methodology. It would be more appropriate to add this information in the methods section.

  • 2.4.3. Pilot intervention (section added to methods)

Page 2, line 88. You mentioned in this line that: “29 pregnant women participating in the pilot intervention”, but in page 10, lines 448-449 you mentioned that 27 pregnant women participated in the pilot intervention. Please check it.

  • Corrected (29)

Page 2, lines 91-92: In page 2, lines 61-62 appears for the first time “revised guideline of Criteria for Reporting the Development and Evaluation 61 of Complex Interventions in healthcare (CReDECI 2)”. Once an acronym is used for a word, this acronym should be used in the rest of the manuscript. Please check it out as this is repeated several times throughout the text.

  • Corrected

Page 3, line 100: 2.1.2. Participants. I think it is a grammar mistake, please check it.

  • Corrected (deleted)

Page 3, line 113: The videos of exercises in digital format are really nice, congratulations.

  • Thank you very much

Page 3, lines 121-128: Was the informed consent obtained from all participants? If so, how was it recorded? Was the study conducted in accordance with the Helsinki declaration? If so, please add this information.

  • Corrected (methods section)

Results

Page 4, lines 196-197. You mentioned that the prenatal exercise program consists of a land-based program to be performed in group. However, due to the pandemic situation, this could be limited. Did you do the sessions online then, but did you all connect at the same time as a group? In my opinion, this information is central to your Please clarify it

  • The pilot intervention was delivered just before the pandemic: “An in-person pilot intervention of 12 sessions over 4 weeks was carried out …. between October 2019 and February 2020” (moved to the methods section, as suggested). However, in Item 2.3 - Description whether exercises are performed individually or in a group: The prenatal exercise program consists of a land-based program to be performed in group. Each session is delivered to groups of 6-12 pregnant women. It is possible to adapt the structure of the group sessions to individual or online sessions, if needed or desired by the participants (e.g., in case of another COVID-19 pandemic…).

Page 5, lines 241-242: The link is not available. Please, check it.

  • https://journals.lww.com/acsmmsse/Fulltext/2004/10000/Development_and_Validation_of_a_Pregnancy_Physical.14.aspx
  • corrected as follows:
  • Development and Validation of a Pregnancy Physical Activity... : Medicine & Science in Sports & Exercise (lww.com)

Page 6, line 268 and 274: You mentioned the Borg scale and the Borg rating of perceived exertion (RPE). Please, add some references.

  • Corrected

Page 9, lines 452-455: In the pilot intervention experience you collected information before and after about satisfaction, adherence, adverse events, types of exercise, and perception of physical fitness improvement. However, you have not collected important information about the baseline situation of the participants. As you mention on page 11, lines 520-532, data must be collected in this baseline situation to later establish a comparative pre-post analysis. In my opinion, you should have included, at least, some of these variables in this study and shown in your results, which would have more conclusively supported the validity of this protocol.

  • The objective of the pilot intervention was to validate the exercise program regarding feasibility and acceptance. Thus, we collected information about satisfaction, adherence, adverse events, types of exercise, and perception of physical fitness improvement, after the intervention. The protocol of a RCT will further describe the health and fitness variables to be collected in baseline and post-intervention. The effectiveness on maternal health and fitness parameters will emerge when the results of our randomized controlled trials are published.
  • This information was clarified in the conclusion section.

Page 11, lines 525-532: In this paragraph appear the main outcome measures to evaluate the effectiveness of the exercise program. Please report some references that support these outcomes, if possible.

  • Item 6 - Description of the control condition (comparator) and reasons for the selection Recent systematic reviews have been showing the effectiveness of physical activity in several maternal health outcomes and in a few fitness parameters [11-14, 24, 25]. However, it is not clear which features are included in the most effective exercise programs, regarding outcomes and long-term adherence.

Discussion

- Your discussion section is in general adequate and complete.

  • Thank you

- You have not described the limitations of your study; please add them before the conclusions section.

  • Section 5. Limitations included, the following statements: The main limitation of the study is that, although the revised guideline CReDECI2 was followed to validate a physical exercise program aimed at improving health and fitness parameters in pregnant women, this process does not guarantee the effectiveness of the intervention. Moreover, the process does not guarantee the absence of obstacles in the design, implementation or evaluation of a future larger‐scale study. It should be noted that during the process of development of the physical exercise program, the hypothesized effects of the intervention were not assessed regarding any of the outcomes of interest highlighted in the literature (i.e., quality of life…gestational diabetes, prevention of falls, depressive symptoms, health-related and skill-related fitness parameters). However, it has been found that, on a consensus basis, experts and potential program users see this type of intervention as relevant and necessary in a healthcare context. This innovative approach of using the CREDECI2 framework to develop an exercise program has made the intervention more likely to be acceptable and deepens understanding of how such a complex intervention performs in primary care, however, the effectiveness of this exercise program in improving maternal health and fitness parameters needs to be evaluated in a RCT. If positive effects are observed, the Active Pregnancy program can be considered for incorporation into the healthcare system. Although this development process enabled us to make a number of improvements to our intervention, and to achieve a design that is likely to be accepted in the setting in which it is to be delivered and tested, avoiding duplicating efforts for subsequent research, the process has some limitations. It was time-consuming, i.e., the process described took 12 months to complete. The pilot intervention included a small number of participants (i.e., less than 30) which may have limited the generalizability of conclusions.

Conclusions

Page 13, lines 629-633: I think conclusions should be rephrased in a more careful way. Your results do not support the statement that “The results of the present work will be useful to assist exercise and health professionals and researchers in…..” You mentioned that it is necessary to develop a study protocol and a pilot study to test the effectiveness of the program.  This should be considered. The results of the study suggested ……

  • Conclusions section corrected, as follows: The CREDECI2 framework for complex interventions provides a structured approach to guide the development of novel interventions in public health. It was extensively followed to validate a prenatal exercise program. The results of the present work can be useful to assist exercise and health professionals, and researchers in planning, promoting, and implementing complex interventions and trials such as a prenatal exercise program. Implications for practice and effectiveness on maternal health and fitness parameters will emerge when the results of our randomized controlled trials are published

I hope that my comments could help to improve the paper.

I would like to congratulate you for all the additional material and YouTube material prepared for this work.

Congratulations for your research.

  • Thank you very much for your time and feedback.
  • We hope that the revisions made in the paper will help to improve it.

Reviewer 5 Report

In this study, the authors report on the validation of a physical exercise program for pregnant women they developed using internationally accepted guidelines (the CERT for development and the CReDECI2 for evaluation of the proposed exercises in pregnancy). I would like to commend the authors for their extensive work and the thoroughness with which they approached the design of the exercises and the production of educational tools. My major objections are that the number of pregnant women changes through the manuscript (first 29, then 27) and that authors often cite various chapters from the 2nd edition of a book that has not yet been published (“Exercise and Physical Activity during Pregnancy and Postpartum”). Furthermore, besides the pregnant women completing the questionnaires and subjectively felt well, the authors should have provided at least some objective (measurable) evidence that their exercise intervention contributed to the well-being of respondents (such as strength measuring or anthropometry before, during and after the 12-week period of exercises). There are many plans on authors` “to do” list.

Abstract

line 19 – please add „The objective of this study was to …“

line 23 – please add comma after „was followed“ and replace „and underwent three stages of“ with „which went through three phases:“

line 28 – here you wrote that 29 pregnant women participated in the study, while on the 10th page, line 448, you wrote 27 women

In case of limited number of words in the abstract, you can exclude abbreviation „(CERT)“ in line 21 because you mention it here only once.

Introduction

line 77 – the first part of this sentence seems to be missing

Materials and Methods

lines 215-216 I do not understand the meaning of what is written here – that recruitment of the target population was planned (together with dissemination and implementation of pilot in gym)?

What was the parity status pregnant women, were all these pregnancies singleton, what were women`s weight and BMI before the pregnancy, what was their weight gain in pregnancy? How were pregnant women recruited: were their pregnancies monitored by gynaecologists/obstetricians who validated the designed exercise program, so these doctors recommended the inclusion of particular pregnant women in the study, or were pregnant women invited to participate in the research through the media? If so, were these women examined by health expert before being included in the intervention or you believed their word that their pregnancies were healthy?

Were these women examined by someone on your team or by one of your associates (MDs) at any time during the pilot exercise study or after 12 weeks of exercises? You wrote that no harmful events were recorded, but it is not clear whether it refers only to the period during the pilot exercises or at all - did you check how their deliveries ended?

Why pregnant women did not sign informed consents?

line 98 – „A report …“, „…with participation in …“

line 99 – „was performed“ please change to „was conducted

line 100 – there is an extra „2.1.2. Participants.“

line 109 – „underwent the three stages“ please change to „was designed through three phases

lines 122-123 – you wrote „No adverse events were reported“; to which period does this sentence refer?

Results

line 138 – what is the reference number of the paper „(Szumilewicz et al., 2022)„?

line 180 – please change to „using 16 items internationally“

line 252 – „WhatsApp

line 261 – change s to c in „three basic sessions“

line 274 – I think you misspelled „RHR“ instead of „HRR“

line 275 – „that corresponds

line 278 - „that corresponds

line 281 – you here for the first time use RM abbreviation, please explain that it means repetition maximum

line 332 – there is an extra space after the full-stop

line 401 - “(Figure 1)”

Lines 421-436 - since your respondents gave answers according to the Likert scale, why did you present their answers as “majority of participants agree or absolutely agree” and not quantitatively?

line 430 – „has potential to be replicated

line 439 „(Figure 2).“

line 448 – there is a discrepancy between the number of participants (here 27, elsewhere 29) - please correct where necessary

line 479 - you wrote: “The protocol of the study will be described elsewhere.” Which protocol do you mean, won`t you continue using the pilot intervention protocol described here?

lines 479-486 - you mentioned many hypothesis here, did you test any of them?

Author Response

Author's Reply to the Review Report (Reviewer 5)

In this study, the authors report on the validation of a physical exercise program for pregnant women they developed using internationally accepted guidelines (the CERT for development and the CReDECI2 for evaluation of the proposed exercises in pregnancy). I would like to commend the authors for their extensive work and the thoroughness with which they approached the design of the exercises and the production of educational tools.

  • Thank you very much for the comment.

My major objections are that the number of pregnant women changes through the manuscript (first 29, then 27)

  • Corrected (N=29)

and that authors often cite various chapters from the 2nd edition of a book that has not yet been published (“Exercise and Physical Activity during Pregnancy and Postpartum”).

  • The book is in press. The first author of the paper is the editor of the book. Those chapters were published in 2019, and then revised in 2022.

Furthermore, besides the pregnant women completing the questionnaires and subjectively felt well, the authors should have provided at least some objective (measurable) evidence that their exercise intervention contributed to the well-being of respondents (such as strength measuring or anthropometry before, during and after the 12-week period of exercises). There are many plans on authors` “to do” list.

  • The objective of the pilot intervention was to check feasibility and acceptance, and not the effectiveness of the program on fitness parameters. A RCT protocol with that purpose is in progress. Nevertheless, more data were added regarding the characteristics of the participants in the methods section (2.4.3. Pilot intervention, Line 193).

Abstract

line 19 – please add „The objective of this study was to …“

  •  

line 23 – please add comma after „was followed“ and replace „and underwent three stages of“ with „which went through three phases:“

  •  

line 28 – here you wrote that 29 pregnant women participated in the study, while on the 10th page, line 448, you wrote 27 women

  • Corrected (N=29)

In case of limited number of words in the abstract, you can exclude abbreviation „(CERT)“ in line 21 because you mention it here only once.

  • OK

Introduction

line 77 – the first part of this sentence seems to be missing

  • Corrected (Fifty participants…). Line 90

Materials and Methods

lines 215-216 I do not understand the meaning of what is written here – that recruitment of the target population was planned (together with dissemination and implementation of pilot in gym)?

What was the parity status pregnant women, were all these pregnancies singleton, what were women`s weight and BMI before the pregnancy, what was their weight gain in pregnancy? How were pregnant women recruited: were their pregnancies monitored by gynaecologists/obstetricians who validated the designed exercise program, so these doctors recommended the inclusion of particular pregnant women in the study, or were pregnant women invited to participate in the research through the media? If so, were these women examined by health expert before being included in the intervention or you believed their word that their pregnancies were healthy?

  • More data regarding the characteristics of the participants were inserted in the methods section (section 2.4.3). All pregnant women were examined by a gynecologist before being included in the pilot intervention. They were recruited by the gynecologists and the exercise physiologists who validated the designed exercise program. This information was added in section 2.4.3. line 193

Were these women examined by someone on your team or by one of your associates (MDs) at any time during the pilot exercise study or after 12 weeks of exercises? You wrote that no harmful events were recorded, but it is not clear whether it refers only to the period during the pilot exercises or at all - did you check how their deliveries ended?

  • All women received standard healthcare, and were clinically examined by gynecologists before, during and after the pilot intervention. All women informed about deliveries and were invited to participate in a postnatal exercise program. This information was added in section 2.4.3. line 193

Why pregnant women did not sign informed consents?

  • They did sign informed consents. This information was added in the methods section (pilot intervention / ethics) lines 193/209

line 98 – „A report …“, „…with participation in …“

  • Corrected

line 99 – „was performed“ please change to „was conducted

  • Corrected

line 100 – there is an extra „2.1.2. Participants.“

  • Deleted

line 109 – „underwent the three stages“ please change to „was designed through three phases

  • Corrected

lines 122-123 – you wrote „No adverse events were reported“; to which period does this sentence refer?

  • No adverse events were reported during the pilot intervention. Corrected

Results

line 138 – what is the reference number of the paper „(Szumilewicz et al., 2022)„?

  • Corrected

line 180 – please change to „using 16 items internationally“

  • Corrected

line 252 – „WhatsApp

  • Corrected

line 261 – change s to c in „three basic sessions“

  • Corrected

line 274 – I think you misspelled „RHR“ instead of „HRR“

  • Corrected

line 275 – „that corresponds

  • Corrected

line 278 - „that corresponds

  • Corrected

line 281 – you here for the first time use RM abbreviation, please explain that it means repetition maximum

  • Corrected

line 332 – there is an extra space after the full-stop

  • Corrected

line 401 - “(Figure 1)”

  • Corrected

Lines 421-436 - since your respondents gave answers according to the Likert scale, why did you present their answers as “majority of participants agree or absolutely agree” and not quantitatively?

  • Quantitative information was added.

line 430 – „has potential to be replicated

  • Corrected

line 439 „(Figure 2).“

  • Corrected

line 448 – there is a discrepancy between the number of participants (here 27, elsewhere 29) - please correct where necessary

  • N=29. Corrected

line 479 - you wrote: “The protocol of the study will be described elsewhere.” Which protocol do you mean, won`t you continue using the pilot intervention protocol described here?

  • The pilot intervention was delivered to test the feasibility of the exercise program. A protocol for a RCT will be developed.

lines 479-486 - you mentioned many hypothesis here, did you test any of them?

  • The protocol of the RCT study will be described elsewhere. Line 492 revised.

  • Thank you very much for your time and feedback.
  • We hope that the revisions made in the paper will help to improve it.

Round 2

Reviewer 2 Report

125-128: “29 healthy pregnant women, with singleton pregnancies….”, I suggest moving to 2.2. section. Also, recommend changing „29” to „Twenty-nine..”

Reviewer 4 Report

IJERPH-1662501_review_R1

Title: ACTIVE PREGNANCY: a physical exercise program promoting fitness and health during pregnancy. Development and validation of a complex intervention

Comments and Suggestions for Authors

Dear authors,

I was glad to have the opportunity to review the new version of your manuscript.

The aim of your study was to build and validate a multicomponent prenatal exercise program to promoting health and fitness during pregnancy, for this, you exhaustively followed the CReDECI2 process.

In my opinion, you have responded positively to the suggestions for improvement made, you have expanded the information required in all the sections, substantially improved the material and methods and discussion sections, reformulated the conclusion and  has also added limitations of the study.

I believe that all these modifications have improved the quality of this manuscript.

Therefore, I congratulate you on your great effort and the work you have done.